# The Effect of Music-Based Rhythmic Auditory Stimulation on Balance and Functional Outcomes after Stroke

**DOI:** 10.3390/healthcare10050899

**Published:** 2022-05-12

**Authors:** Samira Gonzalez-Hoelling, Gloria Reig-Garcia, Carme Bertran-Noguer, Rosa Suñer-Soler

**Affiliations:** 1Neurorehabilitation Department, Hospital Sociosanitari Mutuam Girona, 17007 Girona, Spain; samifisioterapia@gmail.com; 2Department of Nursing, Faculty of Nursing, University of Girona, 17003 Girona, Spain; gloria.reig@udg.edu (G.R.-G.); carme.bertran@udg.edu (C.B.-N.); 3Health and Health Care Research Group, University of Girona, 17003 Girona, Spain

**Keywords:** music therapy, stroke rehabilitation, auditory stimulation, postural balance, functional status

## Abstract

Purpose: the purpose of this paper was to evaluate the effects of music-based rhythmic auditory stimulation on balance and motor function after stroke and whether there are differences depending on the affected hemisphere, lesion site and age. Materials and Methods: This study was an observational and longitudinal study. Adult stroke survivors (*n* = 28), starting no later than 3 weeks after a stroke, conducted 90 min sessions of music-based rhythmic auditory stimulation 3 days a week, in addition to 60 min a day of conventional physiotherapy. Balance ability was evaluated using the Mini Best Test and the Tinetti Test; motor function was evaluated using the Motor Assessment Scale. Results: All of the participants significantly improved their balance ability and motor function variables upon comparing scores at discharge and admission. Intragroup differences were observed upon comparing subgroups of patients by lesion site and by the degree of motor impairment. Age, stroke type and affected hemisphere seemed not to be directly related to the amount of improvement. Conclusions: This study suggests that the effects of music-based rhythmic auditory stimulation (RAS) on balance ability and motor function varies depending on the scale or test used for evaluation and on the variables that the tests measure. Patients with hemiparesis seemed to improve more than those with hemiplegia.

## 1. Introduction

One of the most studied variables in stroke is improvement in walking and, related to this, as a secondary variable, it is common to analyse balance. The most studied type of rehabilitation for balance after stroke is physiotherapy, understood as a mixture of cardiorespiratory endurance and strength work [1,2]. 

Music-based interventions have emerged as a powerful and versatile therapeutic approach in the restoration functional abilities in neurologic conditions. Music-based interventions involves different techniques and devices as active music playing and music listening with music instruments or technical devices. Several clinical studies have proved the effects of music-based interventions to address gait disorders or upper extremity functional abilities in neurorehabilitation [3,4]. Rhythmic auditory stimulation is the use of rhythmic auditory cues, with a metronome or music, in order to synchronise auditory rhythm and motor responses. This entrainment mechanism has an effect on balance and gait patterns in stroke, Parkinson’s disease and other neurological diseases, such as traumatic brain injury [4,5]. The functional and structural architecture of the auditory system is built to rapidly detect patterns of periodicity in acoustic signals [6]. In recent years, Thaut and other authors have studied cerebral adaptations as a result of rhythmic auditory stimuli both in healthy people [7] and people who have suffered a stroke [8]. Most studies cover strokes of the whole of the brain [5] or, at the most, they make a separation between ischaemic and haemorrhagic strokes [9]. There have also been a few studies that compare the effect of RAS on each hemisphere, e.g., studies by Kobinata (2016) [10] and Thaut (2007) [11], which also evaluated the effect by the site of the lesion. Listening to music daily after a middle cerebral artery stroke showed an increase in grey matter volume in frontal and limbic areas both contralaterally and homolaterally to the lesion. This increase was directly related to improvement in cognitive functions and reduced negative mood compared with the condition of patients before listening to music [12]. Despite a dysfunction in the timing mechanism as a result of a stroke with lesions in the cerebellum, thalamus and pons and medulla, RAS has been shown to be effective in facilitating the prediction of motor timing and gait rhythm [10]. The effect on dorsal premotor cortex damage and cerebellar stroke has also been studied [13].

Considering the significant gait improvements after RAS intervention, doubt arises as to whether the same occurs with balance ability. The current evidence regarding the effect of RAS on balance ability after a stroke is highly contradictory. Depending on the scale that is used to analyse the variable, various differing results have been found. Whereas, on the one hand, clinical trials point to significant improvements in balance ability in the RAS group [14,15], meta-analyses and systematic reviews disagree in regard to their interpretation, the quality of the studies and their applicability in improving balance [5,16,17]. 

Although the functional improvements after music-based RAS are clear [16,18], effects by lesion sites and cerebral hemisphere have not been sufficiently studied. This study’s hypothesis was that music-based RAS would prove effective in improving balance ability and motor function post-stroke in our health area and that differences would be observed by clinical and demographical variables. The objective of this study was to evaluate the effects of music-based rhythmic auditory stimulation on balance ability and motor function after stroke and whether there were differences by the affected hemisphere, lesion site and age.

## 2. Materials and Methods

This observational and longitudinal study was a secondary analysis of the intervention group of a controlled clinical trial [19], which was approved by the Bioethical Committee of the hospital and registered before starting at ClinicalTrials.gov (accessed on 21 March 2022) (NCT03974490). This study was conducted and is reported in accordance with the STROBE guidelines.

Inclusion criteria were being 18 years of age or older with a diagnosis of a first stroke (ischaemic or haemorrhagic) or without sequels from a previous stroke, being no more than 3 weeks post-stroke, being a previously independent walker with a Barthel Index > 85, having a hemiparesis with gait alteration after stroke, having a Ranking score of 3–4 or having a Tinetti score < 23. Exclusion criteria were independent walking at admission, having aphasia that impeded communication, having moderate to severe cognitive impairment (Mini-Mental State Examination score < 24), suffering from a musculoskeletal or neurological disorder or not wishing to participate.

### 2.1. Intervention

The intervention is described in detail in our previously published study [19]. The subjects participated three times a week in the rhythmic auditory stimulation sessions, each of which had a duration of 90 min. Furthermore, they received conventional physiotherapy 6 days a week. Conventional physiotherapy consisted of therapeutic exercise and walking training using a parallel walking bar or with assistive devices. Therapeutic exercise was based on proprioceptive neuromuscular facilitation, trunk dissociation, motor control and strengthening exercises. Patients with severe hemiplegia and sensorimotor impairments practiced sitting and standing balance and sit-to-stand using the parallel walking bar. As their physical function improved, they progressed to dynamic standing balance and gait training with assistive devices. Patients began with the intervention on admission to the multidisciplinary intensive rehabilitation unit for subacute stroke, and they finished at discharge; therefore, the number of days in which they participated depended on the length of their hospital stay. 

The rhythmic auditory stimulation sessions consisted of three parts: a 15 min general warm-up; a 60 min main part consisting of rhythmic auditory stimulation and music; and finally, 15 min of stretching and relaxation. The exercises consisted of the multimodal Ronnie Gardiner Method^®^ twice a week and a training session of walking with a metronome and music once a week, including forward walking, sideways walking, military marching, tandem walking, backwards walking and heel and toe walking, through progressive variations and with increases in the speed of the rhythm. The music was chosen by the therapist from different musical genres, both from the past and the present, with a marked 1/4 or 6/8 rhythm and variations in beats per minute from 50 to 110 bpm. The musical pieces for relaxation and stretching were chosen by the participants. The rhythmic auditory stimulation sessions were conducted by a graduate physical and music therapist in the rehabilitation room.

### 2.2. Outcome Measures

Balance ability and motor function were evaluated as dependent variables using the Mini Best Test, the Motor Assessment Scale and the Tinetti Test. 

The Mini Best Test was used for balance ability and consists of 14 sections that make up four of the six areas that the original Best Test evaluates: anticipated postural adjustments, reactive postural control, sensory orientation and dynamic gait. The maximum score is 28 points [20]. The Motor Assessment Scale evaluates the motor function through eight items, including two for leg mobility, three for functional tests of the lower leg (i.e., sitting balance, transfer from sitting to standing walking) and three for the function of the upper limbs. Each item is scored from 0 to 6 [21]. The Tinetti Test is composed of 9 categories related to balance and 10 categories related to gait, scoring from 0 to 28 to evaluate the balance ability of elderly people [22].

In conducting the analysis, age, type of stroke, the hemisphere and lesion site and the degree of motor impairment were taken as independent variables.

Two age groups were used for analysis: under 60 years and 60 years or over, as was done by Ghai et al. (2018) in a systematic review and meta-analysis [23]. The type of stroke was classified, using magnetic resonance imaging (MRI), as either haemorrhagic or ischaemic. Hemispheres were divided into left and right. The lesion site areas were classified into regions innervated by the anterior cerebral artery, regions innervated by the middle cerebral artery, regions innervated by the basal ganglia, vertebrobasilar region, lacunar area and thalamic region. The degree of motor impairment was characterised as hemiparesis (i.e., the patient was mobile and able to withstand resistance) and hemiplegia (i.e., without mobility or unable to withstand resistance). 

### 2.3. Statistical Analysis

Data were analysed using SPSS Statistics version 17.9 (SPSS Inc., Chicago, IL, USA). Continuous variables are described as the mean and standard deviation or the median and interquartile range. Categorical variables are described by the absolute frequency and their percentages. The nonparametric Wilcoxon rank test was used to compare scores at admission and discharge for the Mini Best Test, the Motor Assessment Scale and the Tinetti Scale by type of stroke, the area affected, the hemisphere affected, the degree of motor impairment and age. To study the differences between groups regarding improvement due to the intervention, a new variable was created that measured the difference between the postintervention value and the pre-intervention value for each scale used. The nonparametric Mann–Whitney U test was used to detect differences between two independent samples, and the nonparametric Kruskal–Wallis test was used to detect differences between the scale results by the affected area. In all tests, significance was set at *p* < 0.05.

## 3. Results

In total, 28 people with stroke were included of whom 16 (57.7%) were men and 12 (42.3%) were women. The baseline characteristics of the study participants are shown in Table 1.

All participants significantly improved their balance ability and motor function variables upon comparing scores at discharge and admission (*p* < 0.05), independent of their age, type of stroke and affected hemisphere. Patients with an affected basal ganglia area, middle cerebral area and vertebrobasilar area significantly improved (*p* < 0.05) their Mini Best Test, Motor Assessment Scale and Tinetti Test scores, as shown in Table 2, but patients with impairment of other cerebral areas, such as the anterior cerebral artery and the thalamus, did not significantly improve at discharge compared to admission. In each of these subgroups, there was only one patient; thus, we proceeded to regroup them together for analysis. No significant improvements were obtained for patients with hemiplegia at discharge (*p* > 0.05); on the other hand, patients with hemiparesis significantly improved in all three tests (*p* < 0.05) (Table 2).

To study the difference in improvement between groups due to the intervention, we observed that patients younger than 60 years of age improved as well as those 60 years or older when comparing the difference in the Mini Best Test between admission and discharge (for <60 years, the average rank was 17.33 versus 13.73 for ≥60 years; *p* = 0.340). The Motor Assessment Scale score difference upon admission and discharge did not depend on age (for <60 years, the average rank was 13.33 versus 14.82 for ≥60 years; *p* = 0.694). For the Tinetti Test, the two age groups had the same average rank (i.e., 14.5).

Analysing the results by the type of stroke, there were no significant differences in the average ranges for the Mini Best Test, the Motor Assessment Scale and the Tinetti Test, both at admission and discharge, depending on whether the stroke was haemorrhagic or ischaemic. Participants with haemorrhagic strokes had a slightly better but not significant average rank between admission and discharge on the Mini Best Test compared to patients with ischaemic stroke (15.45 versus 13.97, respectively; *p* = 0.648). Furthermore, individuals with haemorrhagic stroke had a slightly better but not significant average rank in the Motor Assessment Scale than patients with ischaemic stroke (15.90 versus 13.72, respectively; *p* = 0.501), even though the average rank for the Tinetti Test was slightly lower for people with haemorrhagic stroke than for people with ischaemic stroke (13.30 versus 15.70, respectively; *p* = 0.564). 

There were no significant contrasts in the results when comparing the average ranges of the distinct areas in relation to the difference between admission and discharge for the Mini Best Test (average rank: basal ganglia, 16.86; middle cerebral artery, 13.92; vertebrobasilar, 16.06; lacunar, 14.34; other areas, 6.17; *p* = 0.407), the Motor Assessment Scale (average rank: basal ganglia, 15.64; middle cerebral artery, 14.17; vertebrobasilar, 12.13; lacunar, 12.38; other areas, 21.67; *p* = 0.498) and the Tinetti Test (average rank: basal ganglia, 11.43; middle cerebral artery, 14.75; vertebrobasilar, 15.94; lacunar, 18.88; other areas, 11.50; *p* = 0.598).

The analysis did not detect differences among the average ranges of the three tests, either at admission or discharge, depending on whether the stroke had been in the right or left hemisphere (Mini Best Test, *p* = 0.847; Motor Assessment Scale, *p* = 0.943; Tinetti Test, *p* = 0.962).

Patients with hemiparesis significantly improved between admission and discharge on the Motor Assessment Scale compared to patients with hemiplegia (hemiparesis = 18.60 versus hemiplegia = 12.22; *p* = 0.049). No differences were found between patients with hemiparesis and patients with hemiplegia for the Mini Best Test when comparing the difference between admission and discharge. Patients with hemiparesis showed more improvement in the Tinetti Test score than patients with hemiplegia, but the differences were not significant (hemiparesis = 16.06 versus hemiplegia = 11.70; *p* = 0.179).

## 4. Discussion

All participants significantly improved their balance ability and motor function variables upon comparing scores at discharge and admission. However, intragroup differences are observed on comparing subgroups of patients by lesion site and the degree of motor impairment variables. Age, stroke type and affected hemisphere seemed not to be directly related to the amount of improvement.

Upon comparing admission and discharge, patients with haemorrhagic stroke clearly improved their balance ability more when it was measured with the Mini Best Test, whereas patients with ischaemic stroke had slightly higher balance ability average range scores when it was evaluated using the Tinetti Test. As the differences were not significant, the results are not generalisable. However, the results of the three tests were homogenous between patients with ischaemic and haemorrhagic strokes, coinciding with Salvadori (2020). Despite the poor prognosis of haemorrhagic stroke, patients have the same functional recovery as patients with ischaemic stroke [9].

In the present study, patients with basal ganglia lesions were those that most improved statistically, whereas patients with anterior cerebral artery, lacunar and thalamus lesions did not improve and ended up with lower scores than patients with lesions in other areas. It should be pointed out that the number of subjects affected by these lesion sites was low, which could explain the contradiction with other studies in which it has been observed that RAS was effective in improving motor timing and gait rhythm in people with stroke in the cerebellum, pons and medulla and thalamus [10]. Recent studies using neuroimaging have identified neuroanatomical interconnections during treatment with RAS between cortical and subcortical cerebral areas that are distant from one another such as the cerebellum, basal ganglia, thalamus, supplementary motor area, premotor cortex and auditory cortex [6]. Konoike et al. found significant activations, regardless of the body parts, during rhythm perception and reproduction in the inferior frontal gyrus/premotor cortex, inferior parietal lobe and supplemental motor area. The premotor cortex is involved in controlling rhythmic movement, and the cerebellum is essential for controlling discrete movements that require explicit timing control but that are not important for continuous movements [24]. Konoike also concluded that it was difficult to identify the role that each brain region plays in rhythm information processing. Studies of focal lesions in the basal ganglia also provide information about the function of this area in controlling rhythmic movements. Kotz et al. (2011) reported the performance in rhythm perception and reproduction in patients with damage to the basal ganglia due to the fact of stroke, confirming that lesions of the basal ganglia affect the ability to perceive and reproduce rhythm. The basal ganglia reportedly functions by detecting a beat or the metric structure of rhythm and controlling its reproduction [25]. External rhythms may facilitate residual activation of the basal ganglia–cortical circuitry. After 5 weeks of music-based intervention with a group of patients with Parkinson’s disease, Buard et al. (2019) detected significant increases in cortical beta-band activity in neuroimaging results and stronger functional connectivity between auditory and motor areas of the brain [26]. Evidence from Parkinson’s disease patients with implanted neurostimulators in the subthalamic nucleus further demonstrate that rhythmic auditory cues modulate the amplitude of beta oscillations of the subthalamic nucleus during motor performance [27].

No differences were observed in the results at discharge between left-side and right-side strokes, but people with right-side stroke improved their motor function more, whereas people with left-side stroke improved their balance ability more. These results are in line with a study by Särkämö et al. (2014), where it was determined that in both left-side and right-side stroke, there is activation of the grey matter and an increase in its volume in the temporal, frontal, motor, limbic and cerebellar areas, especially contralaterally to the lesion but also in the homolateral hemisphere [12]. Giovanelli (2014) studied the auditory–motor synchronisation of the RAS effect on the premotor dorsal cortex using rTMS, observing that there was no modification of the left side in comparison with the right side. This reinforces the idea that the right dorsal premotor cortex plays a fundamental role in rhythmic auditory–motor synchronisation [13]. The results of Laufer et al. (2003), who compared the balance ability of people with right-side and left-side stroke, are also in contrast to our findings [28]. Grau-Sánchez et al. (2013) observed that motor thresholds are increased in the affected hemisphere, suggesting that cortical neurons increase their thresholds for excitation. Changes in the active motor threshold might explain the specific regulation of the excitability at cortical level rather than spinal [29]. Peyre, in 2020, suggested that restoring the propriospinal reflex to a normal value via listening to music might help to recover locomotor automatism and regain the ability to walk more automatically than without music. This is in contrast to Grau-Sánchez et al. (2013), who found that only one music session modulated the spinal activity involved in stabilised walking, but this was not accompanied by changes in the gait parameters [30].

Given that the Motor Assessment Scale evaluates both functionality and mobility, it was to be expected that people with hemiparesis would have a higher score than those that had hemiplegia and could not move at all [21]. Muscular strength is a predictor of rehabilitation and predisposes to better gait and balance results [9,31], as it was the case in our study with the score differences of the Motor Assessment Scale between patients with hemiparesis and patients with hemiplegia. Reflex motor responses belong to the descending or efferent fibres of the ventral cochlear nucleus that bifurcates bilaterally to the sensory motor tract of the reticular spinal pathway of the spinal cord [32]. When using rhythmic auditory stimulation in task-oriented exercises in hemiparetic extremities, the effects on balance of recruitments between the agonist and the antagonist were shown via electromyography in a study by Tian et al. (2020). Hemiplegic patients in this study had more difficulties in moderating co-contraction, and this may explain the differences between these two subgroups [33]. 

Both age groups (i.e., younger than 60 years and 60 years or older) significantly improved in balance ability and motor function. This coincides with other studies that have found that people over 60 can benefit from a rehabilitation programme using music-based RAS and should not be excluded [4,23,34]. 

Finally, some studies have found contradictory results when comparing the results for balance ability after an RAS intervention when the evaluation was conducted using the Tinetti Test [5], suggesting that this is probably not the best scale to use [35]. There should probably be a more specific scale to determine the differences in relation to evaluating the effect of rhythmic auditory stimulation on functional variables, such as balance ability or motor function.

### Limitations

The size of the sample was a limitation of this study; follow-up studies with larger numbers of participants are likely to provide confirmatory results. Another limitation could be the unequal number of sessions, which was determined by the length of hospitalisation of the patient, although an average of 15 sessions was obtained, in line with other studies [36]. Recent studies show that control of the level of difficulty of each session is more effective than the total number of sessions. As the level of difficulty of the required task increased, the degree of cortical recruitment increased proportionally, and the level of cortical motor activity decreased during the session with repeated exposure. This was due to the temporal information that was stimulated, being passed to subcortical locomotor pathways [31,37].

The strengths of the study are that a multimodal musical intervention that included speech (the Ronnie Gardiner Method^®^) was used and, thus, the Broca area was stimulated. The Broca area is connected with the supplementary motor area, which forms part of the auditory pathway involved in sensorimotor integration, thus increasing the effect [37]. Evaluation using the Mini Best Test and the Motor Assessment Scale are also strong points, as these scales are sensitive and have been validated in both acute and subacute phases after stroke [20,21,38]. 

## 5. Conclusions

This observational and longitudinal study suggests that the use of music-based rhythmic auditory stimulation may affect balance ability and motor function independent of the type of stroke and the affected hemisphere of the brain lesion. The degree of motor impairment at admission is an important outcome factor. The effects varied depending on the scale or test used for evaluation and on the variables that the tests measured. 

## Figures and Tables

**Table 1 healthcare-10-00899-t001:** Baseline characteristics of the study participants.

Variables	*n*(%) or Median (IQR)
Age	66.2 (12.8)
**Sex** Female Male	12 (42.9) 16 (57.1)
**Stroke type** Haemorrhagic Ischaemic	10 (35.7) 18 (64.3)
**Affected side** Right Left	18 (64.4) 10 (35.6)
**Affected area** Basal ganglia MCA Vertebrobasilar Lacunar ACA + MCA ACA Thalamus	7 (25) 6 (21.4) 8 (28.5) 4 (14.3) 1 (3.6) 1 (3.6) 1 (3.6)
**NIHSS** *	5 (3–8)
**Rankin** *	4 (3–4)
**Severity of the impairment** Hemiparesis Hemiplegia	18 (64.3) 10 (35.7)
**Muscular strength at admission** Affected upper limb * Affected lower limb *	4 (2–4) 4 (2.25–4)

The continuous variables are described using the mean and standard deviation or the median and interquartile range and the categorical variables with the absolute frequency and their percentage. * Median (IQR). MCA: middle cerebral artery; ACA: anterior cerebral artery; IQR: interquartile range; NIHSS: National Institutes of Health Stroke Scale.

**Table 2 healthcare-10-00899-t002:** Scores at admission and discharge for the Mini Best Test, the Motor Assessment Scale and the Tinetti Test by age, type of stroke, the area affected, the hemisphere affected and the degree of motor impairment.

Variables	Mini Best Test			Motor Assessment Scale			Tinetti Test		
	Admission	Discharge	*p*	Admission	Discharge	*p*	Admission	Discharge	*p*
**Age group**									
<60 years (*n* = 6)	5.6 (6.1) 2.5 [1.5–13.2]	20.6 (5.6) 21.0 [21.7–28]	0.027	23 (19.5) 26.0 [0–42.5]	39.6 (8.9) 41.0 [30.5–48]	0.028	8.8 (8.2) 8.5 [0–17]	25 (3) 25.5 [21.7–28]	0.027
≥60 years (*n* = 22)	4.1 (6.3) 2 [0–7]	16.5 (5.8) 17.5 [13.7–21]	<0.001	24.1 (16.4) 30 [10–39]	41.9 (9.9) 46 [37.5–47]	<0.001	8.22 (6.5) 8 [1–14]	21.6 (7.6) 24 [20–26]	<0.001
**Type of stroke**									
Haemorrhagic (*n* = 10)	6.5 (8.6) 2.5 [0–14.5]	19.8 (4.8) 10 [15.5–24.2]	0.008	20.6 (17.5) 19 [0–35.2]	39.8 (9.9) 46 [28.7–48]	0.008	9.6 (9) 9 [0–20]	25.1 (3.6) 26 [23.2–28]	0.005
Ischaemic (*n* = 18)	3.4 (4.3) 2 [0–7]	16.1 (6.2) 16.5 [12.7–21]	<0.001	25.6 (16.5) 31 [10–40]	41.2 (9.6) 46 [37.7–47]	<0.001	7.6 (5.3) 8 [1–13]	20.8 (7.9) 24 [20–26]	<0.001
**Affected area**									
Basal ganglia (*n* = 7)	5 (8.9) 2 [0–3]	19 (3.6) 19 [16–22]	0.028	18.7 (16.9) 16 [0–30]	38.2 (11) 46 [28–48]	0.027	7.8 (8) 8 [0–16]	24.8 (3.8) 25 [24–28]	0.018
MCA (*n* = 6)	6.8 (5.1) 9 [2–11.7]	20.5 (5.1) 19.5 [16.2–25.7]	0.028	25.2 (16) 33 [7.5–40.5]	44.1 (6.5) 47 [41.5–47.2]	0.028	12 (4.5) 9 [.7–14.7]	21.8 (9.4) 25.5 [19.2–27.2]	0.028
Vertebrobasilar (*n* = 8)	3.1 (4.6) 1 [0–3]	15.5 (4.79) 16 [12.2–21.7]	0.012	33 (14.3) 38.5 [29.5–42.2]	45.3 (2,6) 46 [44.2–47.7]	0.011	6.2 (5.7) 3.5 [0–12.2]	22 (5.6) 23.5 [18.2–27.5]	0.012
Lacunar (*n* = 4)	5.5 (7.5) 3 [0–13.5]	18.2 (4.35) 17.5 [14.5–22.7]	0.068	30.2 (12) 33.5 [18–39.2]	42.5 (8.3) 46.5 [34–47]	0.144	13 (6.2) 0.5 [0–10.7]	24.7 (2.5) 22.5 [20.2–26.2]	0.068
Other area * (*n* = 3)	0 0	6.3 (6.5) 6 [0–0]	0.180	0 0	24.6 (10) 10 [15]	0.109	12.3 (7.5) 8 [8]	25 (5.1) 28 [19]	0.109
**Affected** **hemisphere**									
Right (*n* = 18)	4.7 (6.7) 2 [0–8]	17.6 (7) 19 [13.7–22.2]	<0.001	23.6 (18.1) 30 [0–41]	41.2 (10.4) 46.5 [37.7–47.2]	<0.000	8 (6.8) 7 [0–14]	21.8 (8.3) 25 [22–27]	<0.001
Left (*n* = 10)	4 (5.3) 2.5 [0–5]	17.1 (3.6) 17 [13.7–19.2]	0.005	24.3 (14.9) 29.5 [11.2–35.5]	39.8 (8.3) 44.5 [29.7–46.2]	0.007	9 (6.9) 9 [1–14.5]	23.2 (3.5) 24 [20.5–25.7]	0.005
**Degree of motor impairment**									
Hemiparesis (*n* = 17)	5.9 (7.3) 21 [14.5–23.5]	19.4 (5) 21 [14.5–23.5]	<0.001	30.2 (16.2) 38 [22.5–41.5]	44.4 (4.8) 46 [44–47.5]	<0.001	9.2 (7.4) 8 [0.5–14]	24.2 (4.6) 26 [21.5–28]	<0.001
Hemiplegia (*n* = 11)	2.5 (3.4) 2 [0–3]	14.2 (6.1) 16 [14–19]	0.180	14.5 (14.1) 13 [0–30]	34.9 (12.3) 30 [24–47]	0.109	7.1 (6) 7 [1–13]	21.3 (7.1) 24 [22–25]	0.109

The continuous variables are described by the mean and standard deviation and median and interquartile range. The nonparametric Wilcoxon rank test was used to compare scores at admission and discharge for the Mini Best Test, the Motor Assessment Scale and the Tinetti Test. MCA: middle cerebral artery; ACA: anterior cerebral artery. * Other areas: We regrouped together anterior cerebral artery, anterior cerebral artery plus middle cerebral artery and thalamus, because there was only one patient in each group.

## Data Availability

Gonzalez, Sami (2020), “Rhythmic Auditory Stimulation after stroke”, Mendeley Data, V1, doi: 10.17632/dsngw3zsnz.1.

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
