# Peer review of "The Effect of Music-Based Rhythmic Auditory Stimulation on Balance and Functional Outcomes after Stroke"

_healthcare, 2022, doi:10.3390/healthcare10050899_

Round 1

Reviewer 1 Report

This study is significant in that it investigated the effect of various variables in stroke patients on the effect of music-based rhythmic auditory stimulation.

However, it is necessary to supplement the following points.

2.1. Intervention

â‘  The effect of walking training will be different depending on the beat of the music and the frequency of the metronome.It is necessary to describe the specific conditions of music used in the training of stroke patients.

2.2. Outcome measures

â‘  Table 3 shows the scores for FIM, Cadence, Gait speed, Upper limb Muscular strength, and Lower limb Muscular strength. However, the method for measuring the above variables was not presented in the 'outcome measures'. A description of the method of measuring the above variables is required.

â‘¡ The evaluation was conducted mainly by the examiner observing the subject. If the functional level and balance ability of the patient were evaluated with objective values using specific equipment, it seems that the patient's ability could be analyzed more effectively.

3. Result

â‘  The data of admission and discharge in Table 3 are not shown separately. It is necessary to organize the data so that it can be displayed separately.

4. Discussion

â‘  In the text from line 175 to line 186, It is the core of this study to explain the reason for the difference in the intervention effect according to each affected area, but the explanation is insufficient in the discussion. Only differences in results from previous studies are presented. Further explanation is needed in this regard.

Author Response

Response to Reviewer 1,

We would like to thank you for your valuable comments.

Comment #1

Intervention: The effect of walking training will be different depending on the beat of the music and the frequency of the metronome.It is necessary to describe the specific conditions of music used in the training of stroke patients.

Reply:

In line 92 we wrote "The intervention is described in detail in our previously published study". However, we have now given more details and added in Lines 115-117 "with a marked pulse, 1/4 or 6/8 rhythm, and variation of beats per minute from 50 bpm to 110 bpm."

Comment #2

Outcome Measure: Table 3 shows the scores for FIM, Cadence, Gait speed, Upper limb Muscular strength, and Lower limb Muscular strength. However, the method for measuring the above variables was not presented in the 'outcome measures'. A description of the method of measuring the above variables is required.

Reply:

We have now removed Table 3 as the outcomes are not directly related to the main aim of the study.

Comment #3

The evaluation was conducted mainly by the examiner observing the subject. If the functional level and balance ability of the patient were evaluated with objective values using specific equipment, it seems that the patient's ability could be analyzed more effectively.

Reply:

In this study we did not use specific equipment such as Biodex, OPTOgait, GAITrite or others gait analysis systems as our aim was not to analyse the balance or gait patterns. Rather we were interested in function, independence in walking, and balance. Furthermore, at the beginning of the study many subjects were unable to stand unaided, and so gait and balance could not have been quantified. However, it was possible to make measurements through observational scales, which are in fact validated for stroke. 

Comment #4

Results: The data of admission and discharge in Table 3 are not shown separately. It is necessary to organize the data so that it can be displayed separately.

Reply:

We have now removed Table 3.

Comment #5

Discussion: In the text from line 175 to line 186, It is the core of this study to explain the reason for the difference in the intervention effect according to each affected area, but the explanation is insufficient in the discussion. Only differences in results from previous studies are presented. Further explanation is needed in this regard.

Reply:

We have now added further explanation:

"Konoike et al. found significant activations, regardless of the body parts, during rhythm perception and reproduction in the inferior frontal gyrus/premotor cortex, inferior parietal lobe, and supplemental motor area. The premotor cortex is involved in controlling rhythmic movement, and the cerebellum is essential for controlling discrete movements that require explicit timing control but which are not important for continuous movements. Konoike also concluded that it was difficult to identify the role that each brain region plays in rhythm information processing. Studies of focal lesions in the basal ganglia also provide information about the function of this area in controlling rhythmic movements. Kotz et al. (2011) reported the performance in rhythm perception and reproduction in patients with damage to the basal ganglia due to stroke, confirming that lesions of the basal ganglia affect the ability to perceive and reproduce rhythm. The basal ganglia reportedly function in detecting beat or the metric structure of rhythm and controlling its reproduction."

Reviewer 2 Report

  1. The title needs revising from “Factors related with the effect of music-based rhythmic auditory stimulation on balance after stroke” to “Musical rhythmic auditory stimulation: Effects on balance after stroke”.
  2. The manuscript needs to be professionally edited by another person to improve readability please. Please confirm this has been done before you resubmit
  3. The title and manuscript have formatting issues throughout – line breaks have been tabbed in when not needed and un-necessary use of hyphens –
  4. In the abstract the statement of purpose needs rephrasing for clarity. Please make it more similar to the revised title above.
  5. The Abstract is hard to follow in places – please especially where you say patients with haemorrhagic stroke had greater improvement in their balance (p= .010) – greater than what or whom?
  6. Likewise where you say patients with ischaemic stroke improved their functional state more (p= .038). More than what or whom please?
  7. In the Abstract please rephrase for clarity: “Those who had right hemi- 18 sphere stroke tended to gain in functional state” as its hard to comprehend
  8. In the Abstract where you say “whereas people with left hemisphere stroke 19 gained in their Tinetti Test scores (p=.034) and in the Mini Best Test (p= .002)” do you mean all of the left hemi strokes showed this and none of the right hemi? (seems hard to believe). Please revise for accuracy
  9. In the Abstract the conclusion does not match up with the results. Please discard the current conclusion and start it afresh with 1 – 2 simple sentences that state clearly the effects of musical cues on movement in stroke.
  10. The Introduction needs major revision for accuracy and expression. Please restructure it to focus on your main aims and hypotheses.
  11. Please remove the inaccurate statement : To date there have been few studies of good methodological quality recording statistically significant improvements with regards to the types of rehabilitation for balance after stroke.
  12. Please also remove the inaccurate statement: It has been demonstrated that physiotherapy, understood as a mixture of cardiorespiratory endurance work and strength work, is currently the best option [1,2]. – this is an over claim
  13. In the Introduction please remove the following un-needed and unclear sentences: “Reflex motor responses belong to the descending or efferent fibres of the ventral cochlear nucleus that bifurcates bilaterally to the sensory motor tract of the reticular spinal pathway of the spinal cord. The functional and structural architecture of the auditory system is built to rapidly detect patterns of periodicity in the acoustic signals [3].”
  14. The following sentence in the Introduction is unclear and can be removed “Despite a dysfunction in the timing mechanism as a result of a stroke with lesions in the cerebellum, thalamus, and pons and medulla, RAS has been shown to be effective in facilitating the prediction of motor timing and the gait rhythm. The effect on dorsa premotor cortex damage and cerebellar stroke has also been studied [11]”.
  15. The following section in the Introduction is unclear and not substantiated with strong evidence and can be removed: Due to the significant gait improvements after RAS intervention, the doubt arises as to whether the same occurs with balance ability.”
  16. The last sentence of the Introduction needs to be revised substantially to make it clear what the exact research question is and the exact aims and hypotheses – linking musical cues with mobility ? by type of stroke?
  17. Methods: Please add a statement about how this observational and longitudinal study is different from the already published controlled clinical trial.
  18. The design is very unclear – please explain what you mean by a longitudinal design.
  19. In 2.1 of the Methods please operationally define physiotherapy
  20. Please rephase the following confusing sentence: “As independent variables, patients were grouped by type of stroke, hemisphere and 108 the lesion site, the degree of motor impairment, and the age of the patients.”
  21. Please add the methods you used for the type of stroke to be classified as either haemorrhagic or ischaemic.
  22. Please rephase for clarity “Patients were grouped 115 by age as under 60 and 60 or over.”
  23. The strategy for statistical analysis is not described in sufficient detail to allow replication. Please have a statistician assist you to re-work this section.
  24. The Discussion needs major revision as currently it is mainly re-stating the results. Please replace the first paragraph with a new one that synthesizes your findings and clarified the impact on stroke rehabilitation.
  25. In the discussion please compare and contrast your findings with other published work on musical cues for neurological patients, including by Jeanette Tamplin, Joanne Wittwer and others. There is an over-reliance on the Thaut paper in this manuscript.
  26. The first sentence of the Conclusion is a major over-claim – your results are not robust enough to make that statement – please write a new more modest sentence
  27. Please remove the last sentence as we know further research is needed.

Author Response

Response to Reviewer 2,

We are grateful for the reviewer’s revision of our manuscript. We provide full replies below to the points that are raised below.

Comment #1

The title needs revising from “Factors related with the effect of music-based rhythmic auditory stimulation on balance after stroke” to “Musical rhythmic auditory stimulation: Effects on balance after stroke”..

Reply:

We have now changed the title to "The effect of music-based rhythmic auditory stimulation on balance and functional outcomes after stroke", which we believe aligns well with the main aim of the study.

Comment #2

The manuscript needs to be professionally edited by another person to improve readability please. Please confirm this has been done before you resubmit

Reply:

In preparing our manuscripts, we always work together with a professional English academic translator. He has now revised the whole manuscript before this resubmission and made a series of changes to improve readability, which are all marked.

Comment #3

The title and manuscript have formatting issues throughout – line breaks have been tabbed in when not needed and un-necessary use of hyphens –

Reply:

Please note that we used the journal’s own template to format the document. We are not quite sure what your concern is with regard to the use of hyphens. We have used hyphens throughout in full compliance with modern English usage, joining, for example, compound adjectives expressing a single idea in this way (e.g. music-based RAS, 90-minute sessions). If the concern is with the use of hyphens at the end of lines to break words between syllables, please note that it is the template that does this.   

Comment #4

In the abstract the statement of purpose needs rephrasing for clarity. Please make it more similar to the revised title above.

Reply:

The purpose in abstract was rephrased and made similar to title: "To evaluate the effects of music-based rhythmic auditory stimulation on balance and functional outcomes after stroke and whether there are differences by the affected hemisphere, lesion site, and age"

Comment #5

The Abstract is hard to follow in places – please especially where you say patients with haemorrhagic stroke had greater improvement in their balance (p= .010) – greater than what or whom?

Reply:

We have rewritten this sentence as follows: "Patients with haemorrhagic stroke had greater improvement in their balance (p= .010) than ischaemic stroke patients whereas this latter group had greater motor function improvement (p= .038)."

Comment #6

Likewise where you say patients with ischaemic stroke improved their functional state more (p= .038). More than what or whom please?

Reply:

The previous reply covers this concern.

Comment #7

In the Abstract please rephrase for clarity: “Those who had right hemisphere stroke tended to gain in functional state” as its hard to comprehend

Reply:

We trust that by changing "functional state" to "motor function" this is now clear.

Comment #8

In the Abstract where you say “whereas people with left hemisphere stroke gained in their Tinetti Test scores (p=.034) and in the Mini Best Test (p= .002)” do you mean all of the left hemi strokes showed this and none of the right hemi? (seems hard to believe). Please revise for accuracy

Reply:

We have rewritten this sentence for greater clarity, explaining that these two tests are related to balance. "Those who had right hemisphere stroke tended to gain in motor function (p= .007), whereas people with left hemisphere stroke gained more in balance with Tinetti Test scores (p=.034) and the Mini Best Test (p= .002)."

Comment #9

In the Abstract the conclusion does not match up with the results. Please discard the current conclusion and start it afresh with 1 – 2 simple sentences that state clearly the effects of musical cues on movement in stroke.

Reply:

As we have now clarified the purpose of the study, the conclusion is now related to the aim of study and follow from the results.  The text of the Conclusions reads as follows:

“This observational and longitudinal study suggests that the use of music-based rhythmic auditory stimulation may affect balance ability and motor function independently of the type of stroke and the site of the brain lesion. The effects vary depending on the scale or test used for evaluation and on the variables that the tests measure.”

Comment #10

The Introduction needs major revision for accuracy and expression. Please restructure it to focus on your main aims and hypotheses.

Reply:

The Introduction, which starts out with more general observations before moving to more specific and detailed ones, has the following structure: the need to study balance after stroke, physiotherapy as the most common therapy in stroke, rhythm perception, studies about the effect of the rhythm on brain after stroke, research about the effect of rhythm depending on brain hemisphere, brain area and stroke type, what has currently not been studied, and the aim of our study.

Having now restated our objective as follows we believe that the Introduction is more focused. We have also now clearly stated our hypothesis. " The study hypothesis is that music-based RAS would prove effective in improving balance and motor function post-stroke in our health area and that differences would be observed by clinical and demographical variables. The objective of the study is to evaluate the effects of music-based rhythmic auditory stimulation on balance and motor function after stroke and whether there are differences by the affected hemisphere, lesion site, and age."

Comment #11

Please remove the inaccurate statement : To date there have been few studies of good methodological quality recording statistically significant improvements with regards to the types of rehabilitation for balance after stroke.

Reply:

We have modified the sentence as follows: "The most studied type of rehabilitation for balance after stroke is physiotherapy, understood as a mixture of cardiorespiratory endurance work and strength work."

Comment #12

Please also remove the inaccurate statement: It has been demonstrated that physiotherapy, understood as a mixture of cardiorespiratory endurance work and strength work, is currently the best option [1,2]. – this is an over claim

Reply:

We have modified the sentence as follows: "The most studied type of rehabilitation for balance after stroke is physiotherapy, understood as a mixture of cardiorespiratory endurance and strength work."

Comment #13

In the Introduction please remove the following un-needed and unclear sentences: “Reflex motor responses belong to the descending or efferent fibres of the ventral cochlear nucleus that bifurcates bilaterally to the sensory motor tract of the reticular spinal pathway of the spinal cord. The functional and structural architecture of the auditory system is built to rapidly detect patterns of periodicity in the acoustic signals [3].”

Reply:

We have removed one sentence as suggested, but we think it is important to keep the sentence about the auditory system and acoustic signals.

Comment #14

The following sentence in the Introduction is unclear and can be removed “Despite a dysfunction in the timing mechanism as a result of a stroke with lesions in the cerebellum, thalamus, and pons and medulla, RAS has been shown to be effective in facilitating the prediction of motor timing and the gait rhythm. The effect on dorsa premotor cortex damage and cerebellar stroke has also been studied [11]”.

Reply:

We think it is important to explain this because of the relationship between different brain areas and rhythm, which is the main aim of the study.

Comment #15

The following section in the Introduction is unclear and not substantiated with strong evidence and can be removed: Due to the significant gait improvements after RAS intervention, the doubt arises as to whether the same occurs with balance ability.”

Reply:

This is an introductory sentence for the paragraph. Evidence is contradictory and different measures are used to assess balance. A Cochrane review by Magee WL and Clark I (2017) expresses reservations with regard to the interpretation of the relevant studies, their quality, and their applicability in improving balance. As a result, it is not possible to make generalisations about the beneficial effects for the time being at least.

Comment #16

The last sentence of the Introduction needs to be revised substantially to make it clear what the exact research question is and the exact aims and hypotheses – linking musical cues with mobility? by type of stroke?

Reply:

As we have mentioned in commentary #10, this has now been done.

Comment #17

Methods: Please add a statement about how this observational and longitudinal study is different from the already published controlled clinical trial.

Reply:

We have now explained that this is a secondary analysis of the intervention group of the previously published clinical trial.

Comment #18

The design is very unclear – please explain what you mean by a longitudinal design.

Reply:

We believe we are using standard terminology here. By a longitudinal design we are referring to the fact that participants were observed and examined to detect changes in outcomes over a period of time.

Comment #19

In 2.1 of the Methods please operationally define physiotherapy

Reply:

Thank you for your suggestion, we have now added the following text:

"Conventional physiotherapy consisted of therapeutic exercise and walking training using a parallel walking bar or assistive devices. Therapeutic exercise was based on proprioceptive neuromuscular facilitation, trunk dissociation, motor control and strengthening exercises. Patients with severe hemiplegia and sensorimotor impairments practiced sitting and standing balance and sit-to-stand using the parallel walking bar. As their physical function improved, they progressed to dynamic standing balance and gait training with assistive devices."

Comment #20

Please rephase the following confusing sentence: “As independent variables, patients were grouped by type of stroke, hemisphere and 108 the lesion site, the degree of motor impairment, and the age of the patients.”

Reply:

We have rewritten this sentence as follows:  “In conducting the analysis, type of stroke, the hemisphere and lesion site, the degree of motor impairment, and age were taken as independent variables.”

Comment #21

Please add the methods you used for the type of stroke to be classified as either haemorrhagic or ischaemic.

Reply:

Classification of the type of stroke was given in Hospital's admission, after magnetic resonance imaging scan (MRI). Therefore, we added in Lines 147-148: “The type of stroke was classified using magnetic resonance imaging (MRI) as either haemorrhagic or ischaemic."

Comment #22

Please rephase for clarity “Patients were grouped by age as under 60 and 60 or over.”

Reply:

We have rewritten the sentence as follows: “Two age groups were used for analysis: under 60 years and 60 years or over.”

Comment #23

The strategy for statistical analysis is not described in sufficient detail to allow replication. Please have a statistician assist you to re-work this section.

Reply:

We did work together with a statistician in writing the original description and we are confident that we have given the appropriate information. We have now added the tests used at the foot of the table 2. Our database is available on reasonable demand.

Comment #24

The Discussion needs major revision as currently it is mainly re-stating the results. Please replace the first paragraph with a new one that synthesizes your findings and clarified the impact on stroke rehabilitation.

Reply:

The first paragraph provides a synthesis of our findings.

In structuring the Discussion we have attempted to discuss our own objectives and results by comparison with the findings of other researchers, which we believe is important to contextualise the work that we are presenting.

We hope that the new text added from lines 230-242 will be considered to improve the quality of this section:

"Konoike et al. found significant activations, regardless of the body parts, during rhythm perception and reproduction in the inferior frontal gyrus/premotor cortex, inferior parietal lobe, and supplemental motor area. The premotor cortex is involved in controlling rhythmic movement, and the cerebellum is essential for controlling discrete movements that require explicit timing control but which are not important for continuous movements. Konoike also concluded that it was difficult to identify the role that each brain region plays in rhythm information processing. Studies of focal lesions in the basal ganglia also provide information about the function of this area in controlling rhythmic movements. Kotz et al. (2011) reported the performance in rhythm perception and reproduction in patients with damage to the basal ganglia due to stroke, confirming that lesions of the basal ganglia affect the ability to perceive and reproduce rhythm. The basal ganglia reportedly function in detecting beat or the metric structure of rhythm and controlling its reproduction."

Comment #25

In the discussion please compare and contrast your findings with other published work on musical cues for neurological patients, including by Jeanette Tamplin, Joanne Wittwer and others. There is an over-reliance on the Thaut paper in this manuscript.

Reply:

We have referred to Thaut and other authors as they have been the principal researchers into the effects of rhythmic auditory stimulation on the brain after stroke with and without diagnostic imaging. Whereas Jeanette Tamplin has published different studies on music-based interventions, they are not related with the effect on the brain after stroke. Even so, we have referred to the systematic review by Magee WL and Clark I (2017) where Tamplin is a co-author. In the case of Joanne Witter, she has published about the effects of rhythm on patients with Alzheimer, Parkinson and other diseases. We have been unable to find any research from this author in which she works with stroke patients.

Comment #26

The first sentence of the Conclusion is a major over-claim – your results are not robust enough to make that statement – please write a new more modest sentence

Reply:

In using the word “suggest” we intended a certain modesty. However, we have now added “may” to make absolutely clear that this is not a question which has been fully resolved yet.

Comment #27

Please remove the last sentence as we know further research is needed.

Reply:

We have now removed this sentence.

Round 2

Reviewer 1 Report

In this study, there is some disappointment in the measurement method to find out function, independence in walking, and balance of stroke patients, but all other parts have been appropriately modified. Thank you for your effort.

Author Response

Reviewer 1

We are grateful for the reviewer’s second review and comments of our manuscript. We hope that you will find the revised version of the manuscript acceptable for publication.

Comment #1

In this study, there is some disappointment in the measurement method to find out function, independence in walking, and balance of stroke patients, but all other parts have been appropriately modified. Thank you for your effort.

Reply:

We are grateful for your second review and comments. Manuscript was revised and terminology of balance ability and motor function was used throughout the document to avoid confusion. Gait or independence in walking was not measured and was not considered as an outcome from the study. Only balance ability and motor function were studied. Three validated test were used for measurement: the Mini Best Test, the Motor Assessment Scale and the Tinetti test.

Table 2 was modified in order to provide more information and make results more clearly.

Language editing services from MPDI author services was used for our manuscript in order to check the overall structure, flow, and clarity of expression.

Reviewer 2 Report

The authors have not addressed in sufficient detail all of my concerns.

  1. In particular the strategy for statistical analysis is not described in sufficient detail to allow replication. Please have a statistician assist you to re-work this section so it is at least twice as detailed.
  2. There is excess self citation
  3. The effects of RAS on conditions other than stroke are not adequately discussed.
  4. There remains excessive repetition in the discussion 
  5. Needs further professional editing for expression and to remove repetition

Author Response

Reviewer 2

We would like to thank you for your comments and review effort.

Comment#1

The authors have not addressed in sufficient detail all of my concerns.

Reply:

We regret your opinion and hope you will be satisfied now with all the changes we did in results and manuscripts structure and flow.

Comment #2

In particular the strategy for statistical analysis is not described in sufficient detail to allow replication. Please have a statistician assist you to re-work this section so it is at least twice as detailed.

Replay:

All statistical methods are now described in the section of statistical analysis and under every table.

We re-worked now to make it replicable as you suggest.

Comment #3

There is excess self-citation

Reply:

We are sorry about your comment and hope the new rewritten and revised manuscript has corrected this point.

Comment #4

The effects of RAS on conditions other than stroke are not adequately discussed.

Reply:

Our study is about the effect of RAS in stroke patients, in our opinion it is not relevant to discuss other clinical application.

We now added in introduction, lines 34-37: "Rhythmic auditory stimulation is the use of rhythmic auditory cues, with metronome or music, in order to synchronize auditory rhythm and motor responses. This entrainment mechanism has effect on balance and gait patterns in stroke, Parkinson disease and other neurological disease as traumatic brain injury".

And in the discussion, lines 214 to 223: " External rhythms may facilitate residual activation of the basal ganglia-cortical circuitry. After 5 weeks of music-based intervention with a group of Parkinson disease patients, Buard et al. (2019) detected significant increase in cortical beta-band activity in neuroimaging results and stronger functional connectivity between auditory and motor areas of the brain. Evidence with Parkinson Disease patients with implanted neurostimulators in the subthalamic nucleus further demonstrated that rhythmic auditory cues modulate the amplitude of beta oscillations of the subthalamic nucleus during motor performance."

Comment #5

There remains excessive repetition in the discussion

Reply:

Discussion was modified in order to discuss more the results. As indicated before, editing services had checked clarity of expression.

Comment #6

Needs further professional editing for expression and to remove repetition

Reply:

We are thankful for your recommendation to use the professional editing to improve our manuscript, this author service was now used.